# Hand Gesture Recognition on a Resource-Limited Interactive Wristband

**DOI:** 10.3390/s21175713

**Published:** 2021-08-25

**Authors:** Shenglin Zhao, Haoyuan Cai, Wenkuan Li, Yaqian Liu, Chunxiu Liu

**Affiliations:** 1State Key Laboratory of Transducer Technology, Aerospace Information Research Institute, Chinese Academy of Sciences, Beijing 100190, China; zhaoshenglin19@mails.ucas.ac.cn (S.Z.); liwenkuan18@mails.ucas.ac.cn (W.L.); liuyaqian20@mails.ucas.ac.cn (Y.L.); cxliu@mail.ie.ac.cn (C.L.); 2University of Chinese Academy of Sciences, Beijing 100086, China

**Keywords:** complementary filter, dynamic time warping (DTW), hand gesture recognition (HGR), inertial measurement unit (IMU), interactive wristband, recurrent neural network (RNN)

## Abstract

Most of the reported hand gesture recognition algorithms require high computational resources, i.e., fast MCU frequency and significant memory, which are highly inapplicable to the cost-effectiveness of consumer electronics products. This paper proposes a hand gesture recognition algorithm running on an interactive wristband, with computational resource requirements as low as Flash < 5 KB, RAM < 1 KB. Firstly, we calculated the three-axis linear acceleration by fusing accelerometer and gyroscope data with a complementary filter. Then, by recording the order of acceleration vectors crossing axes in the world coordinate frame, we defined a new feature code named axis-crossing code. Finally, we set templates for eight hand gestures to recognize new samples. We compared this algorithm’s performance with the widely used dynamic time warping (DTW) algorithm and recurrent neural network (BiLSTM and GRU). The results show that the accuracies of the proposed algorithm and RNNs are higher than DTW and that the time cost of the proposed algorithm is much less than those of DTW and RNNs. The average recognition accuracy is 99.8% on the collected dataset and 97.1% in the actual user-independent case. In general, the proposed algorithm is suitable and competitive in consumer electronics. This work has been volume-produced and patent-granted.

## 1. Introduction

As an intuitive and convenient expression, gesture recognition becomes ubiquitous in human-computer interaction. Gesture recognition technology has been widely used in robot control [1], military tasks [2], authentication systems [3], medical assistance [4], smart home [5], and games [6]. Recently with the rapid development of consumer electronics and the proliferation of technologies, further cutting down computational resources under the premise of ensuring accuracy has become a new requirement. There are mainly two types of gesture recognition methods, i.e., vision-based and inertial sensor-based [7]. Vision-based approaches are subject to ambit, illumination, low sampling rate, and high computational burden. In contrast, inertial sensor-based methods have fewer restrictions when it comes to users’ surrounding environments and relatively lower cost. Therefore, most hand gesture recognition (HGR) studies are based on inertial sensors, especially Micro-Electro-Mechanical System (MEMS) sensors. The technologies to be introduced below are all based on inertial sensors.

Most of the widely-used methods, such as support vector machine (SVM) [5,8,9,10], hidden Markov model (HMM) [11,12,13], neural network [14,15,16,17], and dynamic time warping (DTW) [3,18,19,20,21,22] have achieved good results for recognition accuracy, from 90% to 100%. Traditional machine learning (ML)and neural network (deep learning, DL) methods are data-driven. Their recognition performance depends on whether the training data is sampled adequately for the scene in which they will be used. DTW is considered the best accuracy/computation cost relationship [3] and has been widely applied to speech recognition, gesture recognition, and other signal recognition tasks with time sequence characteristics. However, the main challenge for all above algorithms lies in the misrecognition caused by different preferred speeds and styles, manifested as individual differences, e.g., Y. Wang et al. [10] got the accuracy of recognition at 100% under user-dependent cases, while only 87% accuracy under user-independent cases. This problem can be reduced by expanding the training library for machine learning or selecting appropriate DTW templates adaptively. Besides, the recognition accuracy may drop when adding new gestures to be recognized. Akl et al. [23] tested the recognition accuracy of different algorithms when the number of gesture types increases, where that of classic dynamic time warping (DTW) decreased from 75% to 60% when the types of gestures increased from 12 to 14.

In the cost-focused consumer electronics industry, the requirement of computational resources is a more prominent problem. In terms of computational cost, machine learning methods occupy extensive resources both in the training and recognition stages. DTW may also require a large memory to store the metric matrix, which is always related to the data length. In recent years, some CNN-based deep learning algorithms such as MobileNets [24] and ShuffleNets [25] have emerged to speed up the computing time by reducing the amount of computation. They are both based on depthwise separable convolution, which can theoretically reduce the amount of calculation to one-tenth of the original. When using the CPU for calculation, MobileNets can increase the computing speed by more than three times. However, this level of computing still has relatively high requirements for CPU (Qualcomm Snapdragon 810, for example), which make it difficult to run on a very low-cost MCU, such as Cortex32-M0 as used in this paper. In fact, almost all the studies implement their HGR algorithm on powerful computing devices, such as PC [7,9,10,13,14,18,19,22], smartphone [3,21], or FPGA [15]. The difficulties lie in compromising both on the hardware cost and the algorithm performance. A conventional way for most studies is to proceed to collect inertial sensor data on a cheap microcontroller and transmit them to a PC for the HGR algorithm, e.g., the inertial pen proposed by Hsu et al. in [18,19] and the wrist-worn band proposed by Liu et al. in [22]. In general, the dependence on additional computational equipment presents a cost problem.

For the user’s experience, wired or wireless and hand-held or wearable also need to be considered. Besides integrated devices such as smartphones, most of the studies transmit sensor data to PCs wirelessly [7,9,10,13,18,19,22] or via a data cable [9,14,15]. Some studies developed hand-held modules, such as inertial pen [18,19], sensing mote [13], or HGR for smartphones [3,21]. In most scenarios, wearable devices such as gloves [9] and wristbands [22] are more convenient for users.

A multinational consumer electronics client enterprise demands an interactive wrist band with all the above characteristics, i.e., high recognition accuracy, minimization of commodity cost, and the suitability for wearing. However, the above works do not meet the requirements. Table 1 lists previous studies, detailing their adopted technical solutions, computing hardware, the number of gestures, and the recognition accuracy.

This paper introduces a novel gesture recognition algorithm for a wrist band to interact with intelligent speakers. It neither adopts DTW nor other classic machine learning classifiers and deep learning methods. It adopts a template matching method based on acceleration axis-crossing codes. The significant contributions of this paper include:(a)We introduce a template matching method based on acceleration axis-crossing code and achieved high accuracy in eight gestures both in user-dependent and user-independent cases.(b)The algorithm has a very fast computational speed and can be implemented on resource-limited hardware, which is competitive in consumer electronics.(c)The recognition algorithm does not require an extensive database and does not need to collect as many gestures made by different people as possible to improve the recognition accuracy.


The rest of the paper is organized as follows: Section 2 introduces some related work about hand gesture recognition algorithms using accelerometers and gyroscopes. Section 3 describes our algorithm’s main idea and formulates the model. Section 4 details the implementation process of the whole algorithm. Section 5 compares the proposed algorithm with DTW in terms of accuracy and computational efficiency and provides the accuracy for user-dependent and user-independent cases. Finally, Section 6 concludes our work and discusses possible future research.

## 2. Related Work

DTW is a basic method to calculate the similarity of one-dimensional time-series signals. It ensures a minimum cumulative distance between two aligned sequences and can measure the similarity for the optimal alignment between two temporal sequences [19]. Hsu et al. proposed an inertial pen based on DTW that aligns the trajectories integrated from a quaternion-based complementary filter using accelerometers, gyroscopes, and magnetometers [18,19]. When recognizing eight 3D gestures, they obtained a recognition rate from 82.3% to 98.1% using multiple cross-validation strategies. The inertial pen collects inertial signals on the microcontroller STM32F103TB and transmits them to a PC’s main processor via an RF wireless transceiver for further signal processing and analysis. Srivastava et al. utilized DTW on quaternions and created a quaternion-based dynamic time warping (QDTW) classifier to analyze play styles of a tennis player and provide improvement advice [20]. One of the critical problems of DTW is how to select the class templates in the training stage. Wang et al. selected the minimum intra-class DTW distance as the class template [16]. Hsu et al. then developed a minimal intra-class to maximal inter-class based template selection method as an improvement. There are also other template selection methods based on ML. As a similarity measure, DTW can also be combined with different recognition algorithms [15,23]. Kim et al. replaced the metric calculation of the restricted column energy (RCE) neural network [15] with DTW distance and achieved an accuracy of 98.6%. Akl et al. employed DTW as well as affinity propagation (AP) to improve the training stage [21]. This paper will also train a simple DTW classifier to compare it with the proposed algorithm.

Support vector machine (SVM) [5,8,9,10] and hidden Markov model (HMM) [11,12,13] are two typical machine learning methods in pattern recognition. SVM usually needs careful feature extraction and selection, as well as other traditional ML algorithms like naïve Bayes (NB), K-nearest neighbors (KNN), and decision tree (DT) [16]. HMM is memoryless and unable to use contextual information. In contrast, deep learning (DL) algorithms are becoming a new trend in gesture recognition because they extract and learn hidden features directly from the raw data and usually have higher recognition accuracy. For time sequential signals, recurrent neural networks (RNN) allow information to persist. Thus, we can make full use of the contextual information of the sequence. However, RNN might suffer from a vanishing gradient problem with long data sequences. To solve this problem, Hochreiter et al. designed a special RNN, the long short-term memory (LSTM) network that can learn long dependencies [26]. An LSTM unit is usually composed of a cell, an input gate, an output gate, and a forget gate. The cell remembers values over arbitrary time intervals and the three gates regulate the flow of information into and out of the cell. Ordonez et al. proved that an LSTMRNN better classifies similar gestures than KNN, DT, and SVM [27]. The LSTM was popularized and improved by many researchers. Some variants include bidirectional-long short-term memory (BiLSTM) and gate recurrent unit (GRU) [16,17]. BiLSTM consists of two LSTMs, one taking the input in a forward direction and the other in a backward direction. This structure can effectively increase the amount of information available to the network and improve the context available to the algorithm. As another variant of LSTM, GRU combines the forget gate and the input gate into a single update gate and also combines cell state and hidden state. It has fewer parameters than LSTM. This paper will also train a BiLSTM-RNN and a GRU-RNN as two typical examples of deep learning to compare them with the proposed algorithm.

## 3. Problem Formulation and Modeling

To learn what gesture has been drawn, the most intuitional approach is to restore the spatial space trajectory [18,19]. However, for low-cost MEMS with a high noise ratio, the trajectory obtained by a double integral of the acceleration is always unreliable. The cumulative error will increase seriously when running for a period of time, and thus the integral trajectory is challenging to identify. Therefore, most of the studies utilize the original data or extracted features from accelerometers to develop recognition algorithms. We would also deal with acceleration waveforms. Consider a standard circular motion with a fixed origin in a plane: The relationship between the position vector and time can be modeled as a vector function, shown in Equation (1), where *ω* and *φ* are the certain but unknown angular rate and phase, respectively.
(1)$$p(t)=\left[\begin{array}{c} p_{x}\left(t\right)\\ p_{y}\left(t\right) \end{array}\right]=\left[\begin{array}{c} \cos\left(\omega t+\varphi \right)\\ \sin\left(\omega t+\varphi \right) \end{array}\right]$$

Acceleration is the second derivate of the position, as following: (2)$$a(t)=\left[\begin{array}{c} a_{x}\left(t\right)\\ a_{y}\left(t\right) \end{array}\right]=\left[\begin{array}{c} -\omega^{2}\cos\left(\omega t+\varphi \right)\\ -\omega^{2}\sin\left(\omega t+\varphi \right) \end{array}\right]$$

It can be seen from Equations (1) and (2) that acceleration follows the same circular pattern and position. Representing their directions by tangent angle *θ* as shown in Equation (3), we can conclude that the acceleration direction and the position direction differ by 180°, and their tangent values are equal.
(3)$$\theta_{a}(t)=\tan(\omega t+\varphi +\pi )=\tan(\omega t+\varphi )=\theta_{p}\left(t\right)$$

That gives a revelation that, in some cases, the change of position and acceleration follows the same regular pattern. Further, the evolution of the acceleration vector can directly express the shift in the position vector. We performed a vertical clockwise circle and plotted the accelerations in the world frame. Figure 1 validates that in the YZ-plane, the Y-axis and Z-axis acceleration are two sine waves with about 90° phase difference. Figure 2 dynamically shows the acceleration vector and direction change in a circular gesture. For a circular motion, we can recognize it by recording the angular value of the acceleration vector in the world frame without calculating the trajectory.

For non-circular motions or extremely non-standard circular motions, the above analysis is not directly applicable. We only pay attention to when the quadrant of the vector changes, i.e., when the vector passes through coordinate axes. If considering the angle’s monotonicity, positive and negative of the coordinate axis, there will be eight types of changing vectors. Each pattern is given an identification code, which we call the axis-crossing code. Many gestures can be represented by combining these codes. In this way, although there are differences between the actual trajectory and the measured trajectory, the actual trajectory can still express the characteristics near the coordinate axis through its corresponding acceleration.

## 4. Gesture Recognition Algorithm Based on Axis-Crossing Code

Based on the axis-crossing code mentioned above, a hand gesture recognition algorithm is implemented in this section. It is composed of five procedures: (1) signal acquisition, (2) acceleration coordinate transformation to the world frame, (3) motion mode detection, (4) gesture code generation, and (5) recognition by template codes. The pipeline is shown in Figure 3. The accelerometers and gyroscopes measure signals generated by hand movements in the wristband. For experimental purposes, they can also be sampled by any other inertial modules and collected as a dataset for the PC to perform the evaluation. The other processes will be described in detail below.

### 4.1. Acceleration Coordinate Transformation

This procedure transforms the acceleration from the sensor frame to the earth frame so that the band can be worn on either the left or right hand. Users can gesture with their hands arbitrarily instead of being asked to hold the band flat.

#### 4.1.1. Drift Elimination

The accumulated error of the gyroscope is nonnegligible because it always makes the heading angle drift quickly. The most critical process after acquisition from IMU is to calibrate the random bias of the gyroscope. Ignoring the axis alignment error, the non-orthogonality error, the scale error, and measurement noise (usually assumed Gaussian white noise), a simplified mathematical model of the gyroscope can be expressed as in Equation (4), where **b**_gyro_ is the bias of gyroscope, ***ω*** is the actual value of the triaxial angular velocity, and $\tilde{\omega }$ is the measurement from the gyroscope.
(4)$$\tilde{\omega }=\omega +b_{\mathrm{gyro}}$$

When the normalization of $\tilde{\omega }$ changes minimally and stays smaller than the minimum threshold for a certain length of time, the measurement $\tilde{\omega }$ is considered as the bias **b**_gyro_. By subtracting the bias from the measurement, we can approximately eliminate the drift.

#### 4.1.2. Coordinate Transformation

Accelerometers measure the acceleration aS in the sensor frame (body-fixed frame). It can be expressed with respect to the earth frame by the quaternion qSE and the conjugate q∗SE as shown in Equation (5).
(5)aE=qSE⊗aS⊗q*SE

Eliminating the gravity gE from aE and expressing the rotation defined above in matrix form, we get the transformation equation shown as Equations (6) and (7).
(6)gE=[001]T
(7)aE=RSEaS−gE

Our task in this procedure was to estimate qSE. We utilized the quaternion-based complementary filter (CF) to estimate the rotation information [28,29]. That was done to design a closed-loop control system based on the frequency complementarity characteristics of the accelerometer and gyroscope, to estimate the attitude of the sensor carrier in the form of a quaternion. The implementation is shown in Figure 4.

The controlled error is a cross product of the acceleration measurement a˜S and the estimation a˜S that rotated from the gravity vector by an estimated unit quaternion. $\kappa_{P}$ and $\kappa_{I}$ are proportional and integral gains respectively in the PI controller. Here we design a membership function for $\kappa_{p}$, for the reason that some gestures may bring significant non-gravitational acceleration to the accelerometer, which will distort the error to be controlled calculated by cross product. The membership function is designed as in Equation (8a,b) where τ is the absolute difference between the accelerometer measurement and gravity. It ensures that when the proportion of non-gravitational acceleration increases, the membership degree reduces [30].

(8a)τ=abs(‖a˜S‖−‖gE‖)

(8b)ξ(τ)=exp(−2τ)

Complete iterative equations are shown as follows, where **p**(∙) is the pure quaternion operator, **p**(***ω***) = (0,***ω***).
(9a)e=a˜S×a^S
(9b)$$\delta =\kappa_{P}\xi e+\kappa_{I}\int e$$
(9c)q˙ES=12qES⊗p(ωS+δ)

After the transformation, a low-pass filter should be applied to the linear acceleration a˜E to eliminate the high-frequency component’s influence.
(10)a^kE=αa˜kE+(1−α)a˜k−1E

### 4.2. Motion Detection

This part provides two functional decision processes before calculating the gesture code.

#### 4.2.1. Check Stationary

This function plays the role of segmentation. When the three axes’ accelerations are all less than their respective thresholds for a period, the device is determined to be stationary. At this time, the variables except quaternions are reset to clear the cache for the next gesture to recognize. After a recognition result is given, there is also a timer waiting for clearing variables. The algorithm does not wait for a gesture event to occur and intercepts this whole segment for recognition. The segmentation process and recognition process are carried out simultaneously because gesture codes keep on generating.

#### 4.2.2. Check Shake Gesture

This function is to reset the direction since the definition of clockwise and counterclockwise depends on the observer’s direction. Viewing them from the opposite direction, the clockwise circle and counterclockwise circle are opposite. Therefore, if it starts in an unknown direction, or the heading angle accumulates a significant drift after a long running time, the clockwise circle and counterclockwise circle may be confused. We added a shake gesture to reset the direction. The shake gesture was recognized by detecting the aspect ratio of an acceleration peak. If this value reached a threshold four times continuously, the algorithm determined that a shake gesture occurred. Note that the method of identifying shake gestures here is different from the method to be described below, and thresholds guarantee their distinguishability.

### 4.3. Gesture Code

#### 4.3.1. Projection on Main Plane

Determining the main plane of movement helps to classify gestures in the first step, e.g., vertical or horizontal, and simplifies the two-dimensional plane’s calculation process. Gestures in spatial space can be projected to the vertical or horizontal plane while maintaining similar shapes. We accumulate the linear acceleration amplitudes on the three axes and reset them only when stationary or known gesture duration reaches a threshold. We take the two axes with larger accumulation as the main axis, and the plane determined by them is the main plane. In this way, we have three planes—XY, XZ, and YZ—in the earth frame. The z-axis points vertically up and the x-axis points an uncertain direction horizontally, which is related to the position at the start.

#### 4.3.2. Vector Angle Calculation and Gesture Code Generation

After obtaining the linear acceleration vector and its projection on the principal plane, their quadrant angle *θ* can be defined as in Equation (11).
(11)$$\theta =\{\begin{array}{c} \mathrm{atan}2(Y,X)\;,\;\mathrm{when}\;\mathrm{XY}\;\mathrm{plane}\\ \mathrm{atan}2(Z,Y)\;,\;\mathrm{when}\;\mathrm{YZ}\;\mathrm{plane}\\ \mathrm{atan}2(Z,X)\;,\;\mathrm{when}\;\mathrm{XZ}\;\mathrm{plane} \end{array}$$

We label the four pivotal axes as 1, 2, 3, and 4 in turn and make the following conventions: when the vector passes through the quadrant axis in a clockwise direction, the number of the quadrant axis is taken as a positive label; when the vector passes through the quadrant axis in a counterclockwise direction, the number of the quadrant axis is recorded as a negative label. In this paper, we define four-digit identification codes, i.e., recording the last four labels. Marking the labels as c_1_, c_2_, c_3_ and c_4_, respectively, the calculation rules of the gesture recognition code are as follows:(12)$$c_{\mathrm{ges}}=1000c_{1}+100c_{2}+10c_{3}+c_{4}$$

### 4.4. Recognition by Templates

We made templates for eight gestures: vertical clockwise circle (CWV), vertical counterclockwise circle (CCWV), horizontal clockwise circle (CWH), horizontal counterclockwise circle (CCWH), up (U), down (D), left (L), and right (R). The trajectories of the eight gestures are shown in Figure 5, and the corresponding gesture codes are listed in Table 2.

Figure 6 and Figure 7 show the code templates and corresponding schematic diagrams of circular gestures. See the Appendix A for diagrams corresponding to other identification codes and gestures.

In this way, when the four-digit identification code appears, the algorithm will traverse these code templates to locate the corresponding gesture. Considering users may have inadvertent interference action, each identification code can add the characteristic of duration and amplitude. On the other hand, this may require users to adapt to the execution time and strength of actions. To say the least, an action that is too random cannot be regarded as a gesture. If more types of gestures are requested to be added, we can add new templates to Table 2 or expand the number of digits. But this may result in a decline in the overall accuracy.

## 5. Experiments

The new algorithm presented in this article was evaluated and compared with two typical HGRs —DTW and RNN—using a dataset of 200 × 8 gesture samples. The dataset was collected from an LPMS-B2 module with a sampling rate of 200 Hz. One of the characteristics of this dataset is that the starting point of the circle drawing action is not fixed. It increases the complexity of the waveform of circle gestures and challenges different HGR algorithms. The evaluation and comparison were processed on a PC running the Microsoft Windows 10 operating system with an Intel(R) Core(TM) Processor i7-9700K @ 3.60 GHz, 16-GB RAM, and GPU NVIDIA GeForce RTX 2060 Ti. Then, the proposed algorithm was implemented and tested on the wristband both in user-dependent and user-independent cases. 

### 5.1. DTW Recognizer

The minimum cumulative distance obtained by DTW(∙) programming in Equation (13) represents the similarity of each axis of the two sequences S_i_ and S_j_. The smaller the distance, the higher the similarity between the two sequences. For a three-axis vector, we take its Euclidean distance as the representation of similarity, as shown in Equation (14). Therefore, the most critical task when training a DTW recognizer is to select the optimal DTW template. This article tested three methods to train the template of each gesture:
1.Minimal intra-class (min-intra): This is to find the template sample with the smallest average distance from other samples in each pattern. Equation (15) is a mathematical expression of the above solving process. For a sample S_i_, use DTW to calculate the metric distance with all other samples S_j_ in the same class as the similarity criterion. After traversing all the samples, the sample with the smallest average DTW distance from other samples is selected as the template. 2.Minimal intra-class and maximal inter-class (min-intra and max-inter): The intra-class DTW distance is calculated as the sum of the DTW distance between the template sample and other samples within the same class, while the inter-class DTW distance is calculated as the sum of the distance between the template and all other patterns from the different class [19]. Equation (16) details the specific calculation method, where C_inter_mean_, C_intra_mean_, C_inter_std_, and C_intra_std_ are the means and the standard deviations of the inter-class and intra-class DTW distances, respectively. 3.Maximal inter-class to intra-class (max-inter/intra): For each sample, Equation (17) calculates the average DTW distance between this sample and the inter-class samples divided by the average DTW distance between this sample and the intra-class samples, and take the largest one as the template of this pattern. This is to ensure the maximum difference between classes and within classes.


(13)$$d_{x/y/z}\left(i,j\right)=\mathrm{DTW}(S_{x/y/z,i},S_{x/y/z,j})$$

(14)$$d\left(i,j\right)=\sqrt{d_{x}^{2}\left(i,j\right)+d_{y}^{2}\left(i,j\right)+d_{z}^{2}\left(i,j\right)}$$

(15)$$\underset{i}{\mathrm{argmin}}\frac{1}{N}\sum_{j=1}^{N}d\left(i,j\right),i=1,\ldots ,N$$

(16)$$\underset{i}{\mathrm{argmax}}\left[\left(C_{\mathrm{inter}\_\mathrm{mean}}\left(i\right)-2C_{\mathrm{inter}\_\mathrm{std}}\left(i\right)\right)-\left(C_{\mathrm{intra}\_\mathrm{mean}}\left(i\right)+2C_{\mathrm{intra}\_\mathrm{std}}\left(i\right)\right)\right],\;i=1,\ldots ,N$$

(17)$$\underset{i}{\mathrm{argmax}}\frac{C_{\mathrm{inter}\_\mathrm{mean}}\left(i\right)}{C_{\mathrm{intra}\_\mathrm{mean}}\left(i\right)},\;i=1,\ldots ,N$$

We used all the samples to train the DTW recognizer and tested it with the same data. The recognition accuracy of the three template selection methods is listed in Table 3. The average recognition precision shows that the min-intra and max-inter/intra are similar, and both are better than the min-intra and max-inter methods. This is not in line with our intuition because according to [19], the min-intra and max-inter method are at least not inferior to the min-intra method. We think the reason lies in the complexity of circle gestures due to different starting points and orientations. Figure 8 shows 15 *CWV* gestures, all of which have different starting points, and their orientations are uncertain. We can intuitively see that the waveforms of the same gesture have significant distinction, so the standard deviation of the intra-class distance would be substantial, resulting in the extreme inaccuracy of the min-intra and max-inter method. In fact, the significant distinction between samples within the same class is also the main reason for the low accuracy of the DTW algorithm, no matter which template selection method is used. We chose the min-intra method as the representative of the DTW recognizer to further compare the accuracy and the consumption of computing time.

### 5.2. RNN-BiLSTM and RNN-GRU

This paper referred to the methods in [16] and compared the two most representative and advantageous HGR algorithms of RNN—BiLSTM and GRU—with the proposed algorithm. Their architectures are shown in Figure 9. The RNN-BiLSTM consists of two hidden layers that both have 64 neurons. The RNN-GRU is composed of a first hidden layer of 64 neurons and a second hidden layer of 128 neurons. At the output of the networks, a dense layer (fully connected layer) with eight nodes representing each of the hand gestures and the SoftMax activation function provides the classification probability.

The RNN-BiLSTM and RNN-GRU were implemented using the Keras library. The minibatch approach with a mini-batch size of 64 was used for training. The initial weights of the network were generated randomly. The learning rate was set as 0.001 and the number of training iterations to 40. The trained RNN-BiLSTM and RNN-GRU model classified the data as many-to-one, recognizing the input data as a single label. The inertial data collected from the LPMS-B2 module was divided into 70%(140) as a training dataset and the remaining 30%(60) as a testing dataset. The training process is 40-epochs. Table 4 and Table 5 give the confusion matrix of BiLSTM and GRU, respectively, where the row labels are actual and the column labels are predictions. We can get the average classification accuracy using RNN-BiLSTM of 98.0% and the RNN-GRU of 97.7%. That average classification accuracy of RNN-BiLSTM is better than that of 96.06% using the public data and 94.12% using the collected data reported in [16]. Likewise, the accuracy of RNN-GRU is higher than their 95.34% with their collected data but lower than their 99.16% with the public data. Considering that the data sets used are fundamentally different, it can be assumed that the recognition accuracy achieved by the two articles is similar. 

### 5.3. Comprehensive Comparison with DTW and RNN

This section compares the overall accuracy and time cost of four algorithms on the same PC with an Intel Core processor i7-9700K (3.60 GHz). 

Table 6 presents the comparison of the accuracy of all four HGR algorithms, in which BiLSTM, GRU, and the proposed algorithm all have high recognition accuracy. DTW has the worst accuracy, which has been discussed in Section 5.1. At the same time, the proposed algorithm gets the best accuracy. When considering time consumption, as listed in Table 7, the two RNN algorithms show significant disadvantages. The histogram in Figure 10 provides an intuitive expression of this substantial difference in time cost. It can be seen that among DTW and RNNs, accuracy and time cost compose a trade-off problem: higher accuracy requires more computing time. But our algorithm dramatically reduces the balance point of the trade-off problem. While maintaining high precision, the time cost of our algorithm is only 7.6%, 2.0%, and 3.3% of DTW, BiLSTM, and GRU, respectively.

### 5.4. Implementation and User-Independent Test

We implemented our algorithm on a small circuit board integrated with a Cortex32 M0 chip and IMU module, as shown in Figure 11. The device is as small as the size of two coins and can be fixed on the wrist without external equipment. Figure 12 shows how users interact with the intelligence speaker using our wristband. The user can perform either large or small circle gestures, and the wristband can quickly complete the recognition task online. The recognition result is directly transmitted to the intelligence speaker and its lighting components via Bluetooth to emit different sounds and light colors. The computational resource requirements are as low as Flash < 5 KB, RAM < 1 KB. The maximum battery consumption is about 13 mA × 3.3 V. 

The same gesture performed by different people may vary significantly in speed and amplitude because of their different movement habits. Therefore, it is necessary to test the recognition accuracy for different people to make the product more widely used. We investigated eight participants (six males and two females) to test their individual differences. Before starting the experiment, they were told some points to remember:*1.* *Pause 1–2 s before each gesture to reset the state.**2.* *When drawing a circle, it is better to draw one and a quarter circle or more, and the diameter of the circle should not be too large. The more concise and standard the action is, the higher the accuracy will be.**3.* *When performing straight gestures, it is better to make a short and strong movement without procrastination.*


With these suggestions in mind, the participants were allowed to have some time to get adapted and then they were asked to repeat each gesture 50 times. Each person’s precision is shown in Table 8, and an average precision of 97.1% is achieved. Table 9 shows the confusion matrix in the user-independent case. From Table 5, we conclude a shortcoming that if the user performs the circle gesture too largely, it is easy to be recognized as an up or down gesture. That is why *user2* tested as having a relatively low accuracy comparing to other testers in Table 4. Individual differences could be large if users could not perform each gesture with the necessary consistency.

We further used the false acceptance rate (FAR) and the false rejected rate (FRR) to evaluate the performance of our algorithm in the user-independent case. Table 10 lists the FAR and FRR of each gesture. The results show the average FAR to be 0.44% and the average FRR to be 3.08%. They are close to the authentication system (FAR of 0.27%, FRR of 4.65%) achieved in [3].

## 6. Conclusions and Further Work

This paper proposed and implemented a novel gesture recognition method based on axis-crossing code that can recognize eight gestures. It reached an average recognition accuracy of 99.8% on the collected dataset and 97.1% in the actual user-independent case. The offline test showed that the time consumed by this algorithm was 92.4% less than that of DTW and 97.3% less than RNN (BiLSTM and GRU) on average. It has excellent competitiveness in computing efficiency. Besides, the proposed algorithm dramatically reduces the computational complexity so that gesture recognition can be successfully used on cheap CPUs and MEMS sensors. However, performing large circles may cause serious misidentification, and the individual differences can be relatively significant. It is also challenging to maintain high recognition accuracy when adding new gestures due to the combinations of four-digit axis-crossing codes. The expansion of n-digit codes and combination with deep neural networks will be further studied.

## 7. Patents

This work has a patent granted, CN201910415577. It is available at https://worldwide.espacenet.com/patent/search/family/067490964/publication/CN110109551A?q=CN201910415577 (accessed on 17 May 2019).

Abstract: The invention discloses a gesture recognition method which is applied to the technical field of wearable equipment and comprises the steps of S1 initializing an attitude quaternion of an inertial sensor, a three-axis acceleration vector of the inertial sensor in a geographic coordinate system, a three-axis gyroscope vector of the inertial sensor in a body coordinate system and a gesture feature code; S2, acquiring a three-axis acceleration vector and a three-axis gyroscope vector; S3, based on the three-axis acceleration vector and the three-axis gyroscope vector, judging whether the state of the inertial sensor is a non-static state; S4, calculating and recording a motion vector angle based on the three-axis acceleration vector; S5, judging a gesture identification code based on the motion vector angle; S6, updating the gesture feature code based on the gesture identification code; and S7, searching the gesture feature code in a preset gesture feature code library so as to identify the gesture of the user. The invention further provides a gesture recognition device, an apparatus and a storage medium. According to the present invention, the problem of low recognition precision caused by the higher gesture action requirements in the prior art is effectively solved.

## Figures and Tables

**Figure 1 sensors-21-05713-f001:**
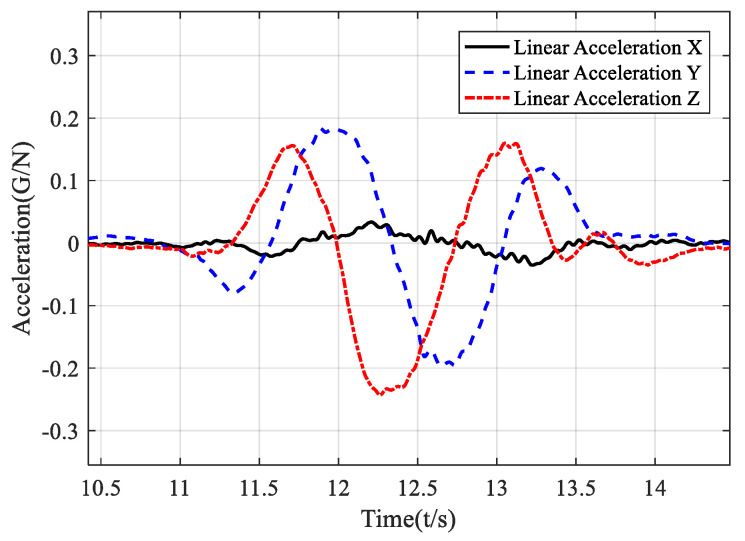
Time domain diagram of acceleration in the local earth coordinates.

**Figure 2 sensors-21-05713-f002:**
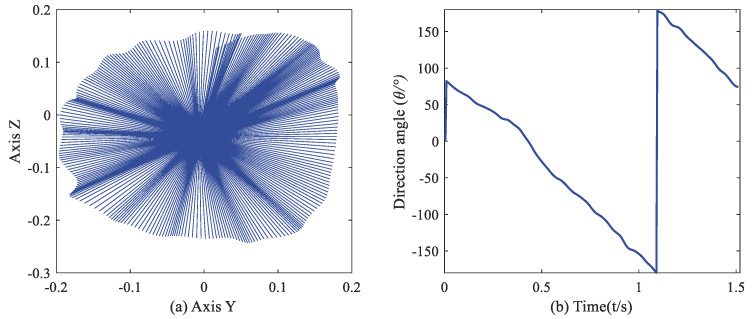
(**a**) Acceleration vector in YZ–plane (in the earth frame). (**b**) Direction angle of acceleration vector varying with time.

**Figure 3 sensors-21-05713-f003:**
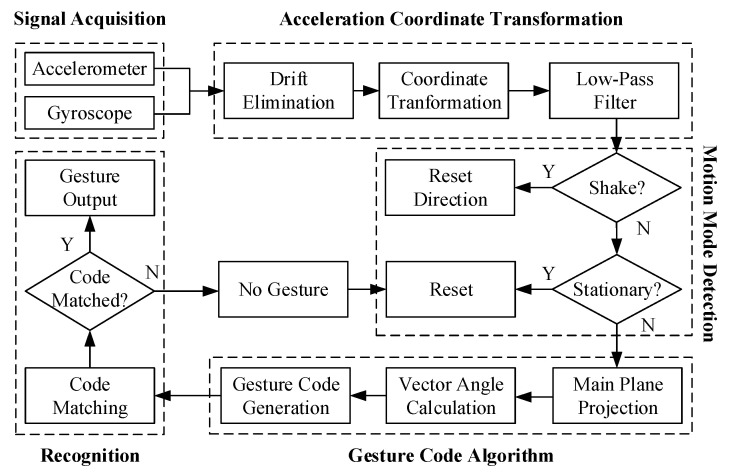
Block diagram of the axis-crossing code based gesture recognition algorithm.

**Figure 4 sensors-21-05713-f004:**
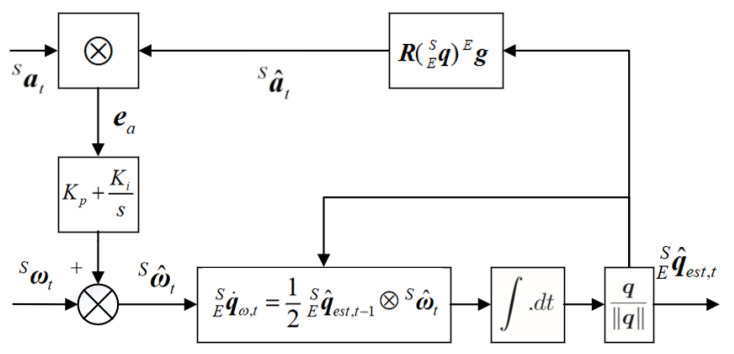
Block diagram of complementary filter for gyroscope and accelerometer.

**Figure 5 sensors-21-05713-f005:**
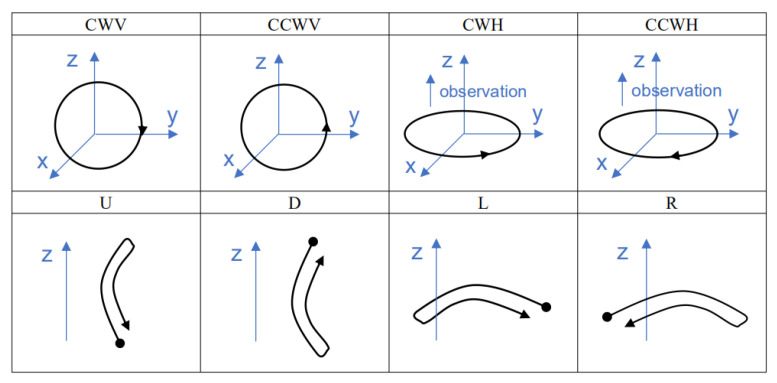
Schematic diagram of eight gestures in spatial coordinate form.

**Figure 6 sensors-21-05713-f006:**
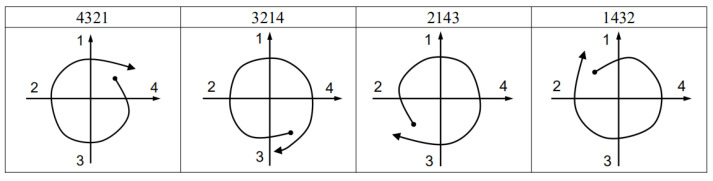
Code templates and schematic diagrams of clockwise circles.

**Figure 7 sensors-21-05713-f007:**
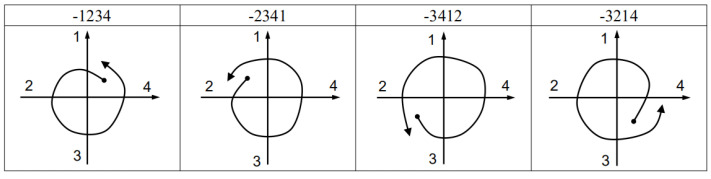
Code templates and schematic diagrams of counterclockwise circles.

**Figure 8 sensors-21-05713-f008:**
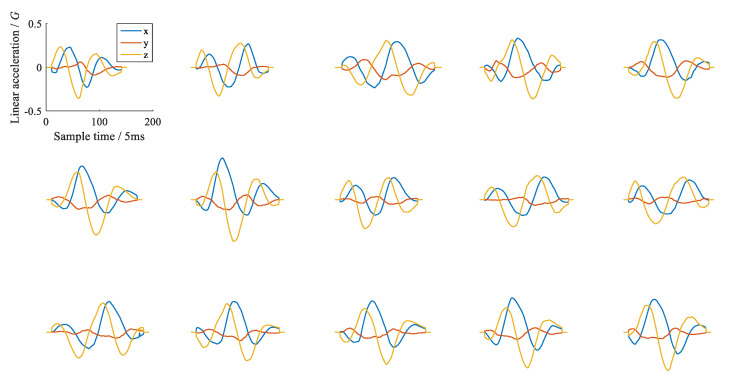
Waveforms of 15 CWV gesture samples whose starting points are unfixed; the waveforms evidently differ.

**Figure 9 sensors-21-05713-f009:**
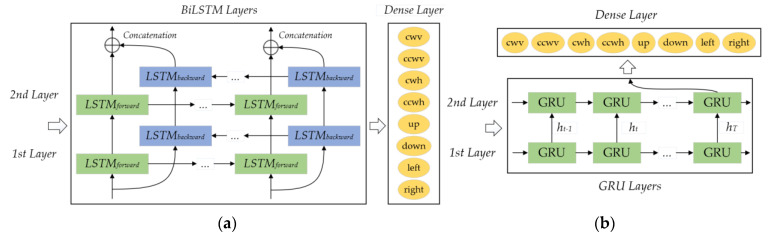
Architectures of (**a**) RNN–BiLSTM and (**b**) RNN–GRU.

**Figure 10 sensors-21-05713-f010:**
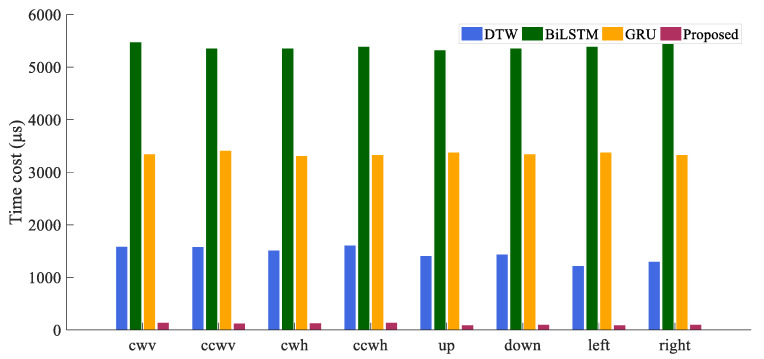
Comparison of time consumed by four algorithms.

**Figure 11 sensors-21-05713-f011:**
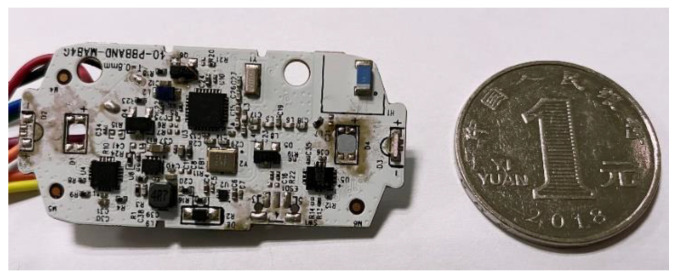
Implemented circuit board and size comparison with a coin.

**Figure 12 sensors-21-05713-f012:**
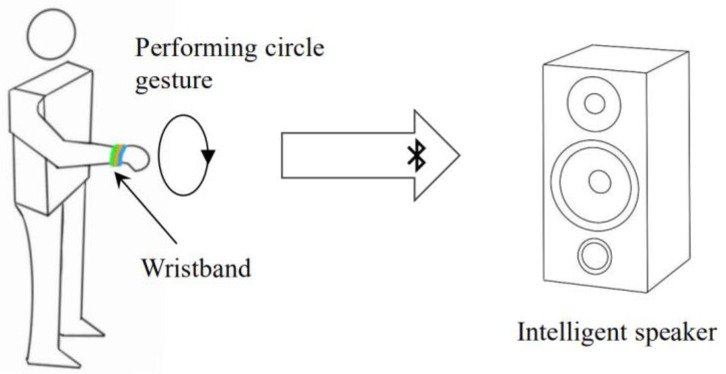
User wearing the wristband interacts with an intelligent speaker.

**Table 1 sensors-21-05713-t001:** Summary of several typical studies.

Device	Solution	Computing Hardware	Number of Gestures	Accuracy
TV controller [5]	MLP/SVM	PIC32MX250F128D	20	98.1%
Sensing module [7]	Sign sequence based method	PC	7	95.6%
Glove [9]	PCA+ANN/ELM/SVM	PC	7	98.1%
Wearable IMU [10]	PCA+LDA+SVM+DTW	PC	10	87.0%
Sensing mote [13]	DCT+HMM	PC	7	95.7%
Pen-style module [14]	FNN+SM	PC	8 (basic)	98.9%
Test platform [15]	RCE neural network+DTW	FPGA	10	98.6%
Patchable IMU [16]	RNN	PC	3	95.3%
Inertial pen [18,19]	DTW	PC	8	98.1%
Continuous HGR [21]	DTW	Smartphones	6	94.0%
Wristband [22]	DTW	PC	12	96.9%
Proposed wristband	Axis-crossing code matching	Cortex32-M0 level MCU	8	97.1%

Computing hardware refers to the platform where the recognition or classification algorithm works. Microcontrollers for signal acquisition and preprocessing are not listed here because they are at the same level as computing. Accuracy is the precision of the 3D hand gesture recognition in the user-independent case. If there was no result, the general recognition accuracy will be displayed. If multiple algorithms are proposed, the table will display the highest accuracy.

**Table 2 sensors-21-05713-t002:** Eight gestures and their code templates.

Gestures	Code Templates
CWV	1432, 4321, 3214, 2143
CWH	1432, 4321, 3214, 2143
CCWV	−1234, −2341, −3412, 4123
CCWH	−1234, −2341, −3412, 4123
U	−2336, 2136, −3357, 1359, 3588, −3568, 1427, 4266, 2659, −3409, −857, −2338, −912, 877
D	−4118, 4318, −409, −4086, 427, −1179, 3177, 1766, −1179, −1786
L	2088, 1359, 4266, 3177, 3209, 4318, 1427, 2136, −2679, −1786, −857, −3568
R	−2268, −3357, −4086, −1179, 877, 1766, 2659, 3588, −4118, −1227, −2336, −3409

**Table 3 sensors-21-05713-t003:** Each gesture’s recognition accuracy of three DTW template selection methods.

	CWV	CWH	CCWV	CCWH	U	D	L	R	Average
**Min-intra**	52.0%	63.5%	54.0%	45.0%	99.5%	38.5%	63.5%	58.5%	59.3%
**Min-intra and max-inter**	34.5%	71.5%	37.0%	68.0%	16.0%	38.5%	20.5%	10.5%	37.0%
**Max-inter/intra**	73.0%	68.5%	71.0%	51.0%	74.0%	51.5%	31.5%	51.0%	58.9%

**Table 4 sensors-21-05713-t004:** Confusion matrix of BiLSTM.

	CWV	CCWV	CWH	CCWH	U	D	L	R
CWV	60							
CCWV		59		1		1		
CWH			59				2	1
CCWH				59			1	2
U		1			60			
D						59		
L			1				57	
R								57

**Table 5 sensors-21-05713-t005:** Confusion matrix of GRU.

	CWV	CCWV	CWH	CCWH	U	D	L	R
CWV	59	1						
CCWV		57		1		1		
CWH			59				1	
CCWH	1			59			2	3
U					60			
D		2				59		
L			1				57	
R								57

**Table 6 sensors-21-05713-t006:** Recognition accuracy of four HGR algorithms for each gesture.

	CWV	CWH	CCWV	CCWH	U	D	L	R	Average
**DTW**	52.0%	63.5%	54.0%	45.0%	99.5%	38.5%	63.5%	58.5%	59.3%
**BiLSTM**	100.0%	98.3%	98.3%	98.3%	100.0%	98.3%	95.0%	95.0%	98.0%
**GRU**	98.3%	95.0%	98.3%	98.3%	100.0%	98.3%	96.7%	95.0%	97.7%
**Proposed**	100.0%	98.5%	100.0%	100.0%	100.0%	100.0%	100.0%	100.0%	99.8%

**Table 7 sensors-21-05713-t007:** Recognition time cost (μs) of four HGR algorithms for each gesture.

	CWV	CWH	CCWV	CCWH	U	D	L	R	Average
**DTW**	1578	1574	1511	1604	1403	1435	1212	1294	1452
**BiLSTM**	5469	5352	5352	5385	5319	5353	5386	5453	5383
**GRU**	3341	3407	3307	3324	3374	3341	3374	3324	3350
**Proposed**	133	123	125	133	86	96	88	95	110

**Table 8 sensors-21-05713-t008:** Accuracy of the user-independent case.

Users	*User1*	*User2*	*User3*	*User4*	*User5*	*User6*	*User7*	*User8*	Average
**Accuracy**	99.5%	92.0%	97.5%	97.4%	98.3%	97.0%	97.3%	97.5%	97.1%

**Table 9 sensors-21-05713-t009:** Confusion matrix of axis-crossing code based recognition algorithm in the user-independent case.

	CWV	CWH	CCWV	CCWH	U	D	L	R
CWV	479				1	4		
CWH	1	491					4	
CCWV			473		4	1		
CCWH				492				7
U	4		4		485	19		
D	16		23		10	476		
L		9					490	2
R				8			6	491

**Table 10 sensors-21-05713-t010:** FAR and FRR of each gesture in the user-independent case.

Gesture	CWV	CWH	CCWV	CCWH	U	D	L	R	Average
**FAR**	0.14%	0.14%	0.14%	0.20%	0.77%	1.40%	0.31%	0.40%	0.44%
**FRR**	4.20%	1.80%	5.40%	1.60%	3.00%	4.80%	2.00%	1.80%	3.08%

## Data Availability

The data presented in this study are available on request from the corresponding author. The data are not publicly available now and will be uploaded later to the public repository.

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
