# Peer review of "Hand Gesture Recognition on a Resource-Limited Interactive Wristband"

_sensors, 2021, doi:10.3390/s21175713_

Round 1

Reviewer 1 Report

The following aspects must addressed:

  • a related work section must be added
  • explain more clearly the novelty of the proposed method
  • papers that perform gesture recognition with accelerometers and gyroscopes must be added in related work section
  • obtained results must be compared with other existing ones (that use data collected from accelerometers and gyroscopes) not only with DTW
  • Section 4.2, figure 11 must be explained more clearly - what is its scope? What about battery consumption?
  • how were selected the 8 gestures that were tested?
  • how can be added more types of gestures? What updates must be added in this case?

Author Response

1. A related work section must be added.

Response: Section 2 Related Work has been added.

2. Explain more clearly the novelty of the proposed method

Response: The main novelty is the axis-crossing gesture code and the complete implementation pipeline, which can significantly reduce computing resources and computing time while achieving high recognition accuracy. Another novelty is based on our previous work that eliminates the drift of yaw angle calculated by the complementary filter. After the transformation of the acceleration vector, the simple recording and matching of vector angle changes according to certain rules can reliably complete the gesture recognition task. Systematic innovations and specific details are described in Section 4.

3. Papers that perform gesture recognition with accelerometers and gyroscopes must be added in related work section

Response: Related work section has included and organized representative papers that combine dynamic time warping (DTW), machine learning, or recurrent neural network (RNN). Of course, they are all based on accelerometers and gyroscopes.

4. Obtained results must be compared with other existing ones (that use data collected from accelerometers and gyroscopes) not only with DTW

Response: We added another comparison with bidirectional-long short-term memory (BiLSTM) and gate recurrent unit (GRU) using inertial sensors. They are two kinds of recurrent neural network (RNN).

The main references include the work in this paper:

Valarezo Añazco, E.; Han, S.J.; Kim, K.; Lopez, P.R.; Kim, T.-S.; Lee, S. Hand Gesture Recognition Using Single Patchable Six-Axis Inertial Measurement Unit via Recurrent Neural Networks. Sensors 2021, 21, 1404, doi:10.3390/s21041404.

5. Section 4.2, figure 11 must be explained more clearly - what is its scope? What about battery consumption?

Response: Does “scope” means the range of action required? We added a sentence in lines 475-477, “The user can perform either large or small circle gestures, and the wristband can quickly complete the recognition task online.” However, over-exaggerated actions may lead to false recognition. Figure 11 has also been updated.

The maximum battery consumption is about 13mA×3.3V. And this sentence is added in line 480.

6. How were selected the 8 gestures that were tested?

Response: It was the company’s demand.

7. How can be added more types of gestures? What updates must be added in this case?

Response: If more types of gestures are requested to be added, we can add new templates to Table 2. This introduction is added in line 338. However, due to the limited combination, we may need to expand the number of digits to increase code combinations. The increase of new types may lead to a decline in accuracy. Therefore, we will need more tests, which may include the adjustment of amplitude, elapse time, and other parameters.

Reviewer 2 Report

. Summary

The authors proposed a pipeline which can efficiently tackle the hand gesture recognition based on sensors. The algorithm neither adopts DTW nor other class machine learning and deep learning methods. Instead, it is developed based on the template matching method. Experiments are conducted compared to the DTW algorithm.

.Strength

-The structure of this paper is appropriate and also the English is not bad.

-Authors developed a working system operates on the embedded board.

. Weakness

-I feel like that presented contents are not technically novel enough to be published in the journal. Many components are heuristically designed (e.g. the definition of templates, gesture code and etc.) and there is no mathematical guarantee for their optimality.

-No state-of-the-art comparison: Authors only compare their algorithm with the algorithm developed based on the DTW operation for the matching.  I think authors need to add more comparison to the SOTA methods (e.g. [15] or more recent one) using larger-scale databases. 

Overall, I think the overall draft needs to be improved a lot before the publication.

Author Response

Weakness1: -I feel like that presented contents are not technically novel enough to be published in the journal. Many components are heuristically designed (e.g. the definition of templates, gesture code and etc.) and there is no mathematical guarantee for their optimality.

Response: The most significant contribution of this paper is that we can significantly reduce computing resources and computing time while maintaining high recognition accuracy. Other algorithms can not meet the requirements of such low computing resources. Therefore, we were also avoiding adopting too complex algorithms. In the off-line test phase, we added the discussion of DTW template training methods and compared the two RNNs, in order to provide more theoretical support.

Weakness2: -No state-of-the-art comparison: Authors only compare their algorithm with the algorithm developed based on the DTW operation for the matching.  I think authors need to add more comparison to the SOTA methods (e.g. [15] or more recent one) using larger-scale databases. 

Response: We added another comparison with bidirectional-long short-term memory (BiLSTM) and gate recurrent unit (GRU). They are two popular kinds of recurrent neural network (RNN) and have been explored in the field of hand gesture recognition.

The main references include this paper: Valarezo Añazco, E.; Han, S.J.; Kim, K.; Lopez, P.R.; Kim, T.-S.; Lee, S. Hand Gesture Recognition Using Single Patchable Six-Axis Inertial Measurement Unit via Recurrent Neural Networks. Sensors 2021, 21, 1404, doi:10.3390/s21041404.

This work is published in 2021, and is one of the SOTA methods.

Reviewer 3 Report

1.This paper proposes a hand gesture recognition algorithm running on an interactive wrist band that installs Accelerometer and Gyroscope. 

2. The designed device efficiently reduces the computation overhead and building cost.

3.The related works clearly exhibit the most related studies providing readers to compare the existing algorithms.

4. Simulation section clearly shows the comparison results.

Author Response

Response:

Thank you for your positive comments!

We added a Related Work section and another comparison with bidirectional-long short-term memory (BiLSTM) and gate recurrent unit (GRU) to present the results more clearly. There are also other modifications, which are marked up using the “Track Changes” function.

Round 2

Reviewer 1 Report

Since all my comments were addressed I recommend to publish the paper.

Author Response

Thank you for your affirmation!

Reviewer 2 Report

After careful reading on the current improved draft and the author responses, I decided to `reconsider' the draft at this stage.

Regarding the technical novelty, I understand what authors are pointing out: avoiding too complex algorithms; however there appears several light-weight deep learning-based approaches including MobileNet arhcitecture and so on. It needs to be clearly stated why the propose approach is better than the stream of works.

Furthermore, I still think authors' insist that they improved the computing resources and time is not clearly demonstrated yet. For newly added results for BiLSTM and GRU, I could not find the description for the exact architecture of BiLSTM and GRU including for example, how many layers are involved in their pipelines which are important factors for both time complexity and accuracy. Also, it's not clear that time complexity is measured on the same device or not. I encourage authors to give more descriptions on this point. Also, [22] is included as the new reference; however its accuracy is not put on any accuracy tables. Please put them on accuracy table and explicitly compare the proposed one with it.

Overall, I think the draft needs to be further improved.

Author Response

All the contents of the last round of modification have been accepted, and this round of modification will be displayed using "revision" function on that basis.
